# Relationships between double cycling and inspiratory effort with diaphragm thickness during the early phase of mechanical ventilation: A prospective observational study

Taiga Itagaki[1]*, Yusuke Akimoto[2], Yuki Nakano[2], Yoshitoyo Ueno[2], Manabu Ishihara[2], Natsuki Tane[3], Yumiko Tsunano[3], Jun Oto[3]

1 Department of Emergency and Disaster Medicine, Tokushima University Hospital, Tokushima, Japan,
2 Department of Emergency and Critical Care Medicine, Tokushima University Hospital, Tokushima, Japan,
3 Department of Emergency and Critical Care Medicine, Tokushima University Graduate School, Tokushima, Japan

* taiga@tokushima-u.ac.jp

**Data Availability Statement:** All relevant data are within the paper and its Supporting Information files.

## Abstract

### Background

Increased and decreased diaphragm thickness during mechanical ventilation is associated with poor outcomes. Some types of patient-ventilator asynchrony theoretically cause myotrauma of the diaphragm. However, the effects of double cycling on structural changes in the diaphragm have not been previously evaluated. Hence, this study aimed to investigate the relationship between double cycling during the early phase of mechanical ventilation and changes in diaphragm thickness, and the involvement of inspiratory effort in the occurrence of double cycling.

### Methods

We evaluated adult patients receiving invasive mechanical ventilation for more than 48 h. The end-expiratory diaphragm thickness ($Tdi_{ee}$) was assessed via ultrasonography on days 1, 2, 3, 5 and 7 after the initiation of mechanical ventilation. Then, the maximum rate of change from day 1 ($\Delta Tdi_{ee}\%$) was evaluated. Concurrently, we recorded esophageal pressure and airway pressure on days 1, 2 and 3 for 1 h during spontaneous breathing. Then, the waveforms were retrospectively analyzed to calculate the incidence of double cycling (double cycling index) and inspiratory esophageal pressure swing ($\Delta P_{es}$). Finally, the correlation between double cycling index as well as $\Delta P_{es}$ and $\Delta Tdi_{ee}\%$ was investigated using linear regression models.

### Results

In total, 19 patients with a median age of 69 (interquartile range: 65–78) years were enrolled in this study, and all received pressure assist-control ventilation. The $Tdi_{ee}$ increased by more than 10% from baseline in nine patients, decreased by more than 10% in nine and

**Funding:** This work was partly supported by JSPS KAKENHI Grant Number JP21K16574 awarded to Dr. Taiga Itagaki. https://kaken.nii.ac.jp/grant/KAKENHI-PROJECT-21K16574/ The funders had no role in study design, data collection and analysis, decision to publish, or preparation of the manuscript. There was no additional external funding received for this study.

**Competing interests:** All authors disclose no conflict of interest.

remained unchanged in one. The double cycling indexes on days 1, 2 and 3 were 2.2%, 1.3% and 4.5%, respectively. There was a linear correlation between the double cycling index on day 3 and $\Delta Tdi_{ee}$% ($R^2 = 0.446$, $p = 0.002$). The double cycling index was correlated with the $\Delta P_{es}$ on days 2 ($R^2 = 0.319$, $p = 0.004$) and 3 ($R^2 = 0.635$, $p < 0.001$).

## Conclusions

Double cycling on the third day of mechanical ventilation was associated with strong inspiratory efforts and, possibly, changes in diaphragm thickness.

## Background

Mechanical ventilation has detrimental effects on the lungs [1]. Moreover, a recent study showed that it can cause diaphragm injury (referred to as ventilator-induced diaphragm dysfunction [VIDD]) [2]. Disuse atrophy owing to the suppression of inspiratory effort and concentric load-induced injury due to contraction against an excessive load are the most important causes of VIDD [3–5]. Patients receiving mechanical ventilation commonly present with either increased or decreased diaphragm thickness. Both conditions are associated with prolonged ventilator dependence and a higher mortality rate in patients admitted to the intensive care units (ICU) of hospitals [6, 7].

Some types of patient-ventilator asynchrony, including premature cycling, ineffective efforts and reverse triggering theoretically cause myotrauma of the diaphragm due to eccentric load-induced injury [3–5]. Eccentric contractions during muscular lengthening lead to more injuries compared with concentric contractions [8]. Meanwhile, the effects of double cycling, the most lung-injurious type of patient-ventilator asynchrony [9, 10] on VIDD have not been evaluated. Double cycling is characterized by two consecutive ventilator cycles separated by a short expiration [11]; therefore, the diaphragm may be forced to contract eccentrically without ventilatory assistance during the short expiration. In addition, excessive load on the diaphragm causes muscle injury (concentric load-induced injury) [3, 12, 13]; thus, double cycling may also be associated with diaphragm injury because it frequently occurs if the ventilatory demand is high [14, 15]. Thille et al. emphasized the possible deleterious effects of asynchronies, including double cycling, which are correlated with increased energy expenditure and an abnormal diaphragmatic pattern [14]. However, the effects of double cycling on diaphragm function have never been investigated.

There is a recommendation to lower the sedation level of mechanically ventilated patients as soon as their condition stabilizes because of the physiological benefits of spontaneous breathing [16–18]. According to the largest international epidemiological study of acute respiratory distress syndrome (ARDS) [19], about 30% of patients were managed in spontaneous breathing mode from the first day of mechanical ventilation. However, the early awakening strategy may possibly cause strong inspiratory efforts that adversely affect the lungs [20] and diaphragm [3], especially in ARDS patients Thus monitoring of inspiratory effort is important when we reduce sedation [21].

We hypothesized that double cycling is associated with increased diaphragm thickness caused by strong inspiratory effort. To test this hypothesis, we investigated the relationship between double cycling during the early phase of mechanical ventilation and changes in diaphragm thickness. Additionally, we examined the association of changes in inspiratory effort over time with the occurrence of double cycling.

## Methods

This prospective observational study was conducted in the ICU of Tokushima University Hospital. Moreover, it was approved by the ethics committee of the institution (protocol number: 3273). Written informed consent was obtained from the families or guardians. This research was preliminarily registered as a clinical trial (UMIN clinical trial registry: 000033533). We recruited study participants during from September 1, 2018, to June 1, 2020, and follow-ups of all patients ended on June 10, 2020.

### Participants and their management

We assessed adult patients within 24 h after receiving invasive mechanical ventilation, which was anticipated to be continued for more than 48 h. The exclusion criteria were as follows: patients aged below 18 years, with trauma or chest tube insertion at the measurement point, receiving treatment with continuous neuromuscular blocking agent (NMBA) infusion, and diagnosed with esophageal disease.

Throughout the study period, all patients received pressure assist-control ventilation using the same ICU ventilator (PB840 or PB980) (Covidien, Mansfield, Massachusetts). The inspiratory pressure was set to obtain a tidal volume of 6–8 mL/kg ideal body weight. Parameters such as positive end-expiratory pressure, fraction of inspired oxygen, respiratory frequency, inspiratory time and flow trigger sensitivity were adjusted by bedside physicians. Other patient management strategies were performed by the bedside physicians and nurses according to the critical care guidelines and the analgesia-sedation protocol of our institution.

### Diaphragm thickness

On days 1, 2, 3, 5 and 7 after the initiation of mechanical ventilation, the diaphragm thickness at peak inspiration and end-expiration ($Tdi_{ee}$) were examined via ultrasonography during spontaneous breathing under pressure assist-control ventilation. On days 1, 2, and 3, esophageal pressure ($P_{es}$) was examined; the patient was assumed to be spontaneously breathing when the deflection of $P_{es}$ was initiated before the rise of airway pressure ($P_{aw}$). On days 3 and 7, the patient was assumed to be spontaneously breathing when the actual respiratory frequency was greater than the set value for respiratory rate. We assessed the percentage of change from the baseline $Tdi_{ee}$ ($\Delta Tdi_{ee}\%$). Patients were classified into three groups (increased, decreased and unchanged diaphragm thickness groups) based on the maximum $\Delta Tdi_{ee}\%$ using a 10% cutoff value according to previous studies [7, 22]. For each measurement, diaphragm thickness was calculated as follows:

Thickening fraction (%) = [(thickness at peak inspiration–thickness at end-expiration) / thickness at end-expiration] × 100

Data collection was discontinued in case of extubation, discharge from the ICU, or death, whichever occurred first. The procedures for measurement were discussed in-depth in our previous study [23]. To summarize, measurement was performed using B mode ultrasonography with a liner transducer perpendicularly placed on the right chest wall at the zone of apposition [24]. Each recording was performed by the same investigator. Then, the actual measurement of diaphragm thickness was retrospectively conducted by the same investigator using stored images to blind the data analysis from patient status.

### Incidence of double cycling and inspiratory efforts

After inclusion of patients in the study, a feeding tube equipped with an esophageal balloon catheter (SmartCath, Vyaire Medical, Mettawa, Illinois) was placed nasally through the

stomach to facilitate the continuous measurement of $P_{es}$. $P_{es}$ was evaluated using a pressure transducer (TM6600, San-You Technology, Saitama, Japan). Correct balloon placement was confirmed using the occlusion technique as described in a previous study [25]. Flow and $P_{aw}$ were assessed. The pneumotachometer (model 3700A, Hans-Rudolph, Shawnee, Kansas) and pressure transducer (TM6600, San-You Technology, Saitama, Japan) were placed between the heat and moisture exchanger and Y-piece of the breathing circuit. The pneumotachometer was connected to a differential pressure transducer (TP-602T, Nihon-Kohden, Tokyo, Japan). All signals were amplified, sent to an analog/digital converter with a sampling rate of 100 Hz, then analyzed with a data-acquisition software (WINDAQ, Dataq Instruments, Akron, Ohio). We recorded $P_{es}$, $P_{aw,}$ and flow simultaneously for 1 h on days 1, 2 and 3 if the patients were spontaneously breathing (all assisted), thereby assuring negative $P_{es}$ swing preceding the start of insufflation.

We retrospectively analyzed the waveforms to calculate the incidence of double cycling (double cycling index) and inspiratory esophageal pressure swing ($\Delta P_{es}$). Double cycling was defined as two ventilator-delivered cycles separated by an extremely short expiratory time occurring within a single inspiratory effort [14]. The double cycling index was calculated as follows:

Double cycling index (%) = [number of double cycling / total number of esophageal pressure waveforms representing spontaneous breaths] × 100

To calculate the mean $\Delta P_{es}$, we collected 10 consecutive stable esophageal pressure waveforms unaccompanied by double cycling. Then, the last three breaths were analyzed.

## Endpoints of the study

The primary endpoint of this study was the correlation between the double cycling index on the first 3 days and the maximum $\Delta Tdi_{ee}$. The secondary endpoints were changes in the mean end-expiratory diaphragm thickness over time during mechanical ventilation, the correlation of the double cycling index with $\Delta P_{es}$ during the first 3 days, and the effect of sedation levels on changes in inspiratory effort.

## Clinical data collection

Data regarding demographic characteristics, acute physiology and chronic health evaluation (APACHE) II score upon ICU admission, cause of mechanical ventilation and clinical outcomes were collected from the medical charts. Within the first 3 days of mechanical ventilation, we calculated the time-weighted averages of the following ventilatory variables: driving pressure, positive end-expiratory pressure, flow trigger sensitivity, inspiratory time, and respiratory rate. Moreover, the average tidal volume of 3–5 representative normally triggered and double cycled breaths was obtained by integrating the flow-time waveforms. In addition, we assessed the time-weighted Richmond Agitation–Sedation Scale (RASS) score and fentanyl use on days 1, 2 and 3.

## Statistical analysis

Continuous data were presented as medians with interquartile range (IQR), and categorical variables as numbers and percentages. The differences between continuous variables were assessed using $t$-test or Mann–Whitney U test. Categorical variables were presented as appropriate using the chi-square test or the Fisher's exact test.

Linear regression models were user to assess the correlation of the double cycling index on the first 3 days with the maximum $\Delta Tdi_{ee}$ or $\Delta P_{es}$. To assess effects over time, changes in $Tdi_{ee}$, double cycling index, $\Delta P_{es}$, thickening fraction, RASS, and fentanyl use were analyzed via one-

way analysis of variance (ANOVA) to reveal statistically significant differences between at least two time points. Afterwards, we conducted Tukey's HSD test for multiple comparisons.

All analyses were carried out with the Statistical Package for the Social Sciences software version 26 (SPSS Inc., Chicago, Illinois). A p value of $< 0.05$ was considered statistically significant.

## Results

In total, 27 patients who met the inclusion criteria were enrolled in this study. However, 8 patients were subsequently excluded due to equipment failure (n = 4) and lack of consent form (n = 4). Finally, 19 patients were included in the analysis (Fig 1). The median (IQR) age was 69 (65–78) years, and the median APACHE II score was 27 (24–30). Among the patients, 14 (74%) were men. Acute hypoxemic respiratory failure (58%) was the most common cause of mechanical ventilation. All patients received pressure assist-control ventilation. Table 1 shows the detailed characteristics and clinical outcomes of each group. Clinical outcomes were not significantly different among patients with increased and decreased diaphragm thickness.

Diaphragm thickness was evaluated in 100%, 100%, 100%, 58%, and 28% of patients on days 1, 2, 3, 5 and 7, respectively. Within the first 7 days after mechanical ventilation, the $Tdi_{ee}$ increased by more than 10% from baseline (difference: +13.7%; 95% confidence interval [CI]:

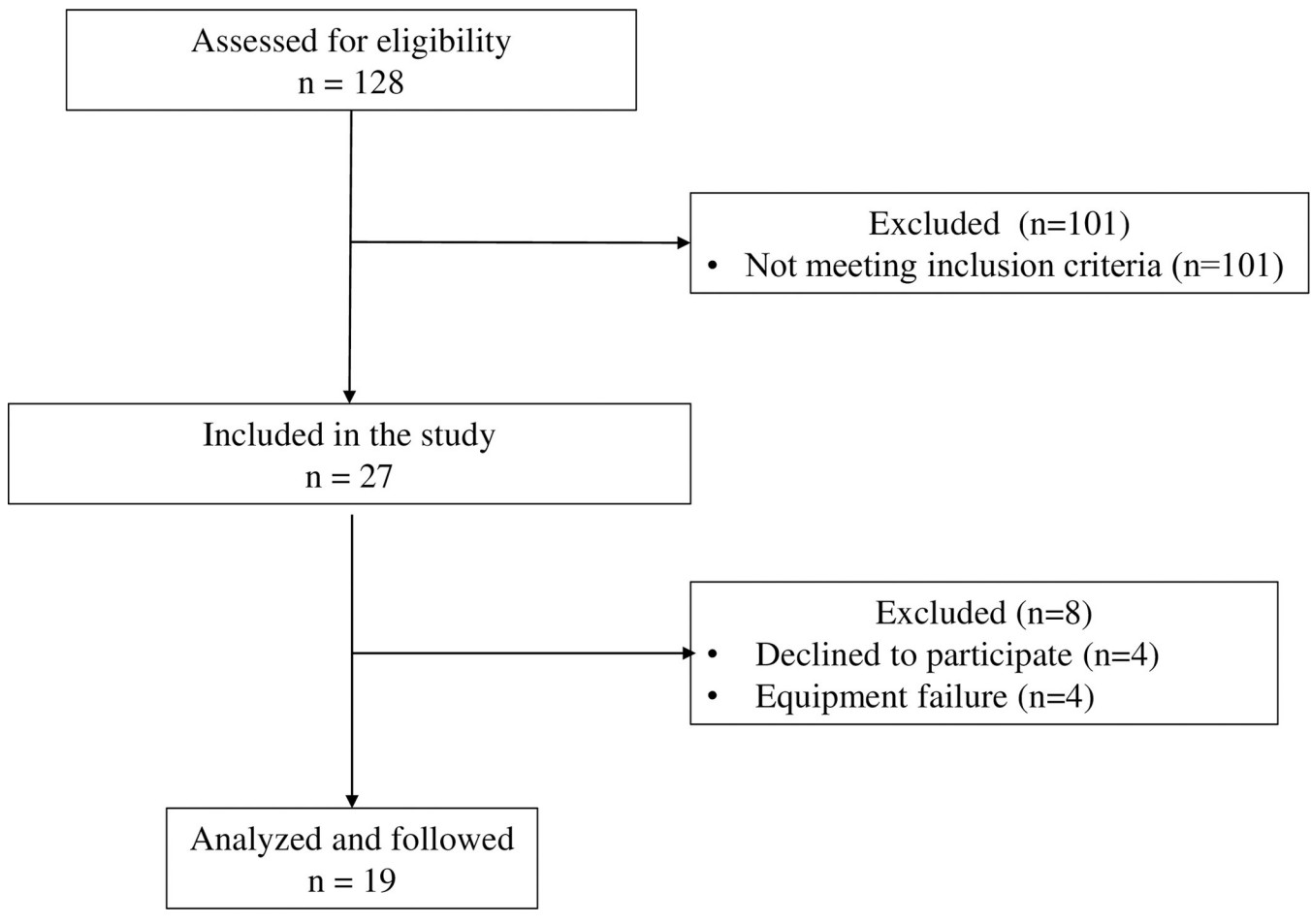

**Fig 1. Flowchart of study participants.**

**Table 1. Patient characteristics and clinical outcomes.**

| | Increased (n = 9) | Decreased (n = 9) | Unchanged (n = 1) | P value |
|---|---|---|---|---|
| Age, yr | 70 (67–80) | 69 (65–82) | 38 | 0.604 |
| Male sex, n (%) | 1 (89) | 6 (67) | 0 (0) | 0.576 |
| Height, cm | 164 (162–169) | 157 (155–170) | 160 | 0.475 |
| Weight, kg | 59 (50–65) | 66 (55–74) | 50 | 0.157 |
| APACHE II score | 27 (32–38) | 28 (25–31) | 8 | 0.636 |
| Reason for mechanical ventilation, n (%) | | | | |
| Acute hypoxemic respiratory failure | 6 (67) | 5 (56) | 0 (0) | 1.000 |
| Decompensated heart failure | 2 (22) | 1 (11) | 0 (0) | 1.000 |
| Consciousness disturbances | 0 (0) | 1 (11) | 1 (100) | 1.000 |
| Others | 1 (11) | 2 (22) | 0 (0) | 1.000 |
| Clinical outcomes | | | | |
| Ventilator free days within 28 days | 20 (0–23) | 20 (10–22) | 0 | 0.396 |
| Duration of ICU stay, days | 8 (6–11) | 10 (7–16) | 6 | 0.404 |
| Reintubation, n (%) | 2 (22) | 3 (33) | 0 (0) | 1.000 |
| Tracheostomy, n (%) | 0 (0) | 2 (22) | 0 (0) | 0.471 |
| ICU mortality, n (%) | 3 (33) | 0 (0) | 1 (100) | 0.206 |

Data are expressed as median with interquartile range unless otherwise noted. P values indicate comparisons between increased (n = 9) versus decreased (n = 9) diaphragm thickness. APACHE, Acute physiology and chronic health evaluation; ICU, intensive care unit.

+6.5%–+21.0%; p = 0.036) in 9 patients, decreased by more than 10% (difference: −13.9%; 95% CI: −18.0% to −9.8%; p < 0.001) in 9 patients and remained unchanged in 1 patient (Fig 2). The maximum $\Delta Tdi_{ee}$% was +31.4% (+17.1%–+42.5%) in the increased diaphragm thickness group, −24.0% (−34.1%–−20.3%) in the decreased diaphragm thickness group and +9.8% in the unchanged diaphragm thickness group.

Table 2 shows the variables of double cycling, inspiratory effort, mechanical ventilation, and sedation status within the first 3 days. The double cycling indices on days 1, 2, and 3 were not significantly different between the increased and decreased diaphragm thickness groups. $\Delta P_{es}$, but not thickening fraction, which is a measure of inspiratory effort, was significantly higher in the increased diaphragm thickness group than in the decreased diaphragm thickness group on days 2 and 3 (p = 0.002 and 0.0026, respectively). There was no statistically significant intergroup difference in terms of ventilator settings and tidal volumes of both normally triggered and double cycled breaths within the first 3 days. RASS score and fentanyl use were also not significantly different between patients with increased and decreased diaphragm thickness groups on day 3. However, sedation was more likely to be lightened over time in the increased diaphragm thickness group (p = 0.069, ANOVA) even though fentanyl use did not change.

The maximum $\Delta Tdi_{ee}$% did not correlate with the double cycling index on days 1 and 2, but these had a linear correlation ($R^2$ = 0.446, p = 0.002) on day 3 (Fig 3). Fig 4 shows the relationship between the double cycling index and the $\Delta P_{es}$ on days 1, 2 and 3. The double cycling index correlated with $\Delta P_{es}$ on days 2 ($R^2$ = 0.319, p = 0.004) and 3 ($R^2$ = 0.635, p < 0.001).

## Discussion

The two most important findings of this study are as follows: First, the double cycling on day 3 correlated with the maximum changes in end-expiratory diaphragm thickness during

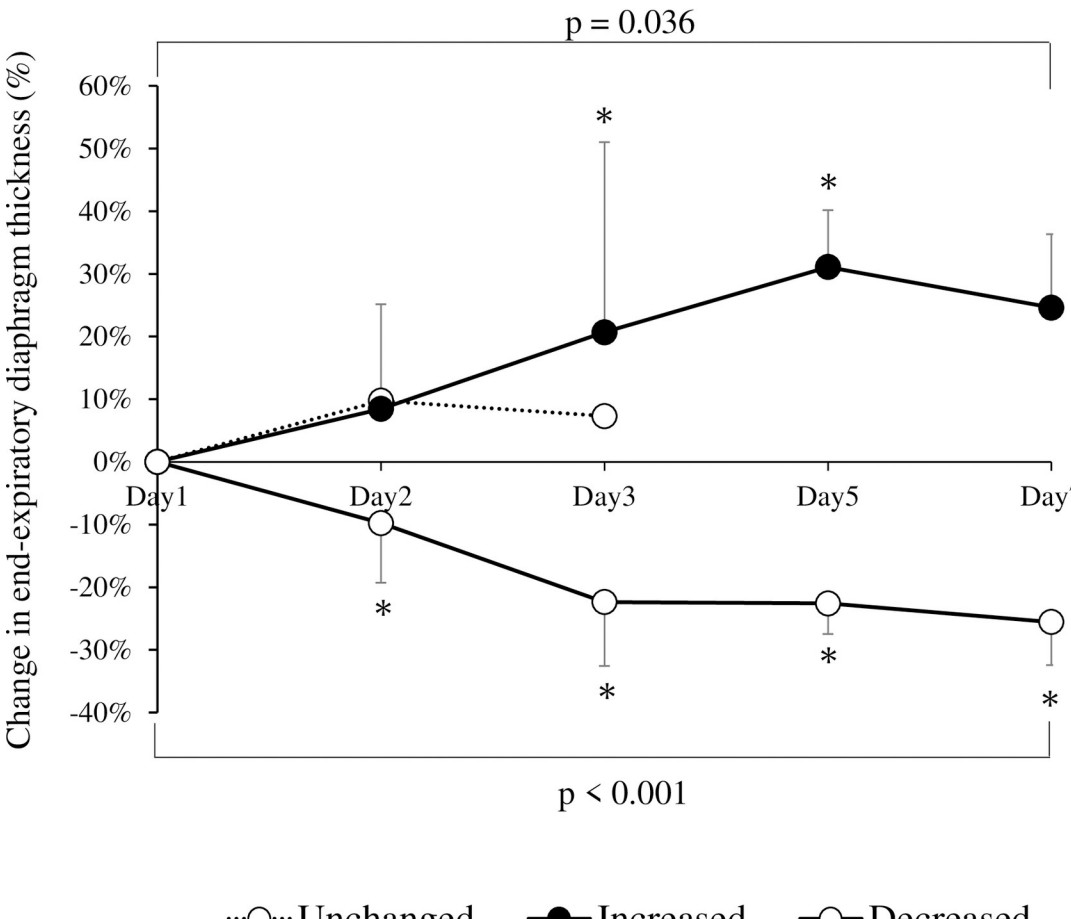

**Fig 2. Changes in mean end-expiratory diaphragm thickness over time during mechanical ventilation in each group.** The error bars indicate 95% confidence intervals. p values represent statistically significant differences between at least two time points (analysis of variance). *p < 0.05 versus day 1.

mechanical ventilation. Moreover, the incidence of double cycling was associated with inspiratory efforts on days 2 and 3.

This is the first study to show the association of double cycling with increased diaphragm thickness, which possibly resulted in diaphragm dysfunction. Our hypothesis that double cycling is associated with the changes in diaphragm thickness due to strong inspiratory effort turned out to be plausible. However, eccentric contraction of the diaphragm, a possible mechanism by which a certain type of asynchronies affects the diaphragm [8], and the characteristics of double cycling itself, such as increased tidal volume [11], may have still influenced diaphragm injury. Further studies are needed to clarify this issue.

Disuse atrophy and concentric load-induced injury are the common mechanisms of diaphragm dysfunction during mechanical ventilation [3–5]. Concentric load-induced injury is a type of acute diaphragm injury leading to hypertrophy caused by excessive diaphragm contraction when ventilatory support is insufficient against inspiratory effort [4, 12, 13]. Excessive inspiratory effort is also a risk factor of double cycling [14, 15]. In fact, inspiratory effort has been positively correlated with the occurrence of double cycling. Thus, in situations where strong inspiratory effort is present, both double cycling and concentric load-induced injury are likely to occur.

**Table 2. Variables of double cycling, inspiratory effort, ventilation, and sedation status within the first 3 days of mechanical ventilation.**

| | Increased (n = 9) | Decreased (n = 9) | Unchanged (n = 1) | P value |
|---|---|---|---|---|
| Double cycling index, % | | | | |
| Day 1 | 1.1 (0.2–3.8) | 0.2 (0.0–1.2) | 0.1 | 0.225 |
| Day 2 | 1.5 (0.4–2.0) | 0.4 (0.2–0.4) | 0.2 | 0.207 |
| Day 3 | 5.7 (2.2–8.3) | 0.4 (0.0–2.3) | 0.3 | 0.100 |
| $\Delta P_{es}$, cmH$_2$O | | | | |
| Day 1 | −7.4 (−12.1−−2.9) | −2.9 (−5.1−−0.7) | −1.0 | 0.068 |
| Day 2 | −6.5 (−11.1−−3.1) | −2.6 (−4.9−−1.6) | −0.8 | 0.022 |
| Day 3 | −12.2 (-16.3−−7.5) | −2.9 (−8.3−−1.2) | −2.5 | 0.026 |
| Thickening fraction, % | | | | |
| Day 1 | 14.6 (3.2–42.6) | 17.5 (11.5–21.5) | 17.1 | 0.394 |
| Day 2 | 11.6 (5.9–31.3) | 14.6 (6.1–21.1) | 31.1 | 0.357 |
| Day 3 | 18.8 (11.0–38.0) | 13.2 (8.3–17.4) | 26.1 | 0.086 |
| Ventilatory variables over the first 3 days | | | | |
| Driving pressure, cmH$_2$O | 12 (10–14) | 12 (10–14) | 12 | 1.000 |
| PEEP, cmH$_2$O | 10 (8–11) | 10 (7–11) | 8 | 0.793 |
| Flow trigger, L/min | 3.0 (2.5–3.0) | 3.0 (3.0–3.0) | 3.0 | 0.169 |
| Inspiratory time, sec | 0.9 (0.8–1.1) | 1.0 (0.9–1.0) | 1.2 | 0.948 |
| Respiratory rate, per min | 18 (12–22) | 16 (15–20) | 15 | 0.800 |
| $V_T$ of normally triggered breath, mL/kg IBW | 9.2 (8.1–11.3) | 9.2 (6.9–10.4) | 7.8 | 0.577 |
| $V_T$ of double cycled breath, mL/kg IBW | 14.3 (8.8–16.4) | 11.1 (8.9–14.7) | 10.1 | 0.638 |
| RASS | | | | |
| Day 1 | −3.0 (−3.5−−1.8) | −2.5 (−3.5−−1.7) | −5 | 0.595 |
| Day 2 | −2.5 (−3.8−−1.0) | −2.5 (−3.5−−1.0) | −4 | 0.821 |
| Day 3 | −2.0 (−2.5−−0.5)* | −2.0 (−3.8−−1.0) | −2 | 0.150 |
| Fentanyl use, μg/kg | | | | |
| Day 1 | 12.5 (8.9–14.2) | 9.8 (5.4–12.2) | 12.0 | 0.770 |
| Day 2 | 8.5 (6.1–10.8) | 4.3 (3.5–8.9)* | 0.8 | 0.867 |
| Day 3 | 8.5 (4.4–11.3) | 4.3 (0.7–6.9)* | 0.0 | 0.310 |

Data are expressed as median with interquartile range unless otherwise noted. P values indicate comparisons between increased (n = 9) versus decreased (n = 9) diaphragm thickness.

*p < 0.05 versus Day 1. $\Delta P_{es}$, inspiratory esophageal pressure swing; PEEP, positive end-expiratory pressure; $V_T$, tidal volume; IBW, ideal body weight; RASS, Richmond Agitation-Sedation Scale.

In this study, inspiratory effort increased leading to day 3 of mechanical ventilation. This phenomenon might be attributed to the escalation of patient awakening according to our analgesia-first sedation protocol, since we observed a lower RASS score, and fentanyl use remained higher on day 3 in patients with increased diaphragm thickness. Preventing deep sedation and maintaining spontaneous breathing have been the gold standard of care for patients receiving mechanical ventilation [26], because maintaining spontaneous breathing has several physiological benefits including the prevention of diaphragm atrophy [16–18]. However, strong inspiratory effort on the diaphragm and lungs, known as concentric load-induced injury [4, 12, 13] and patient self-inflicted lung injury (P-SILI) [20], particularly in patients with severe ARDS, have detrimental effects. Whether increased inspiratory efforts and diaphragm thickness affected clinical outcomes in our patients remains unclear. Nevertheless, the continuous use of the analgesia-first sedation protocol might have led to unfavorable consequences. In particular, fentanyl use could have increased respiratory drive and eventually caused double

## Double cycling index on day 3

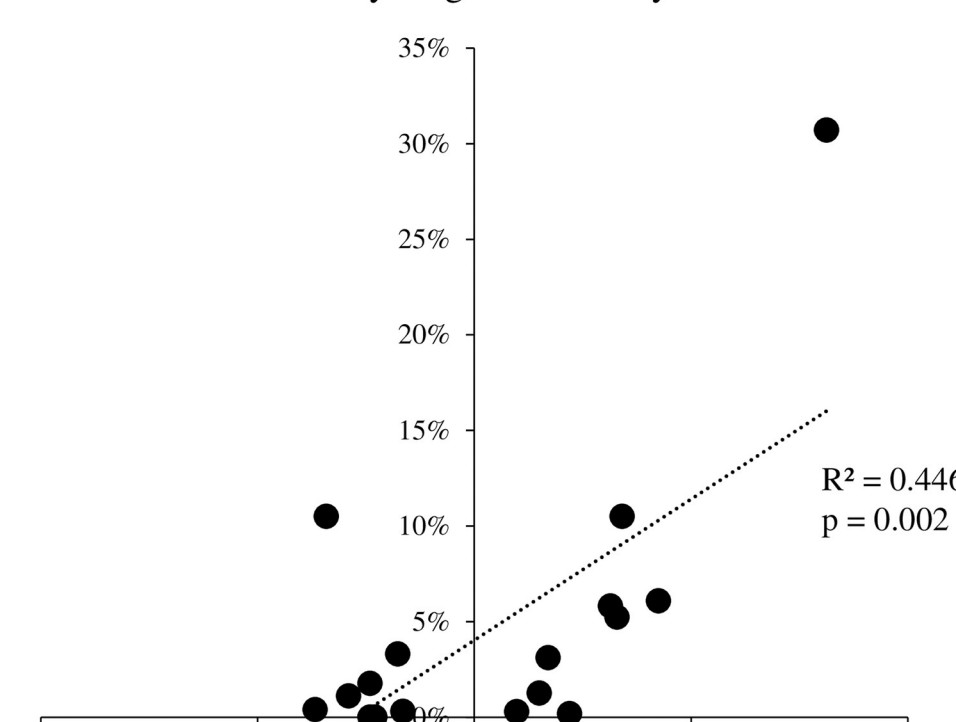

**Maximum change in end-expiratory diaphragm thickness**

**Fig 3. Correlation between the maximum change in end-expiratory diaphragm thickness from baseline and the double cycling index on day 3.**

cycling. Ferguson and Drummond assessed the acute effects of fentanyl on breathing patterns, revealing that the durations of inspiration and expiration increased by 30% and 95%, respectively. Furthermore, the tidal volume was elevated in proportion to inspiratory duration [27]. Opioids can provide light and comfortable sedation [28]. However, caution should be taken in patients with increased inspiratory efforts.

This study had several limitations. First, the analysis only included 19 adult patients with and without respiratory failure. Thus, neither we could extrapolate our findings directly to other patients nor perform multivariate logistic regression to discover whether double cycling was independently associated with diaphragm thickness. Moreover, one patient was presented with diaphragm atrophy (maximum $\Delta Tdi_{ee}$%: −34.1%) despite a high double cycling index (10.5%). Although we could not determine the cause of atrophy, future studies with a larger sample size can clarify this issue. Second, the effect of double cycling itself on the diaphragm, such as increased tidal volume caused by double cycling, or the effect of double cycling unaccompanied by strong inspiratory effort were not investigated. We observed that both increased and decreased diaphragm thickness groups tended to have increased tidal volumes of double cycled breaths, but this was not statistically significant. In addition, minimal double cycling was observed in patients without increased inspiratory efforts. Thus, the effect of double cycling caused by inadequate ventilator settings (e.g., extremely short inspiratory time and low tidal volume [15, 29]) on the diaphragm is unclear. Third, only the pressure assist-control

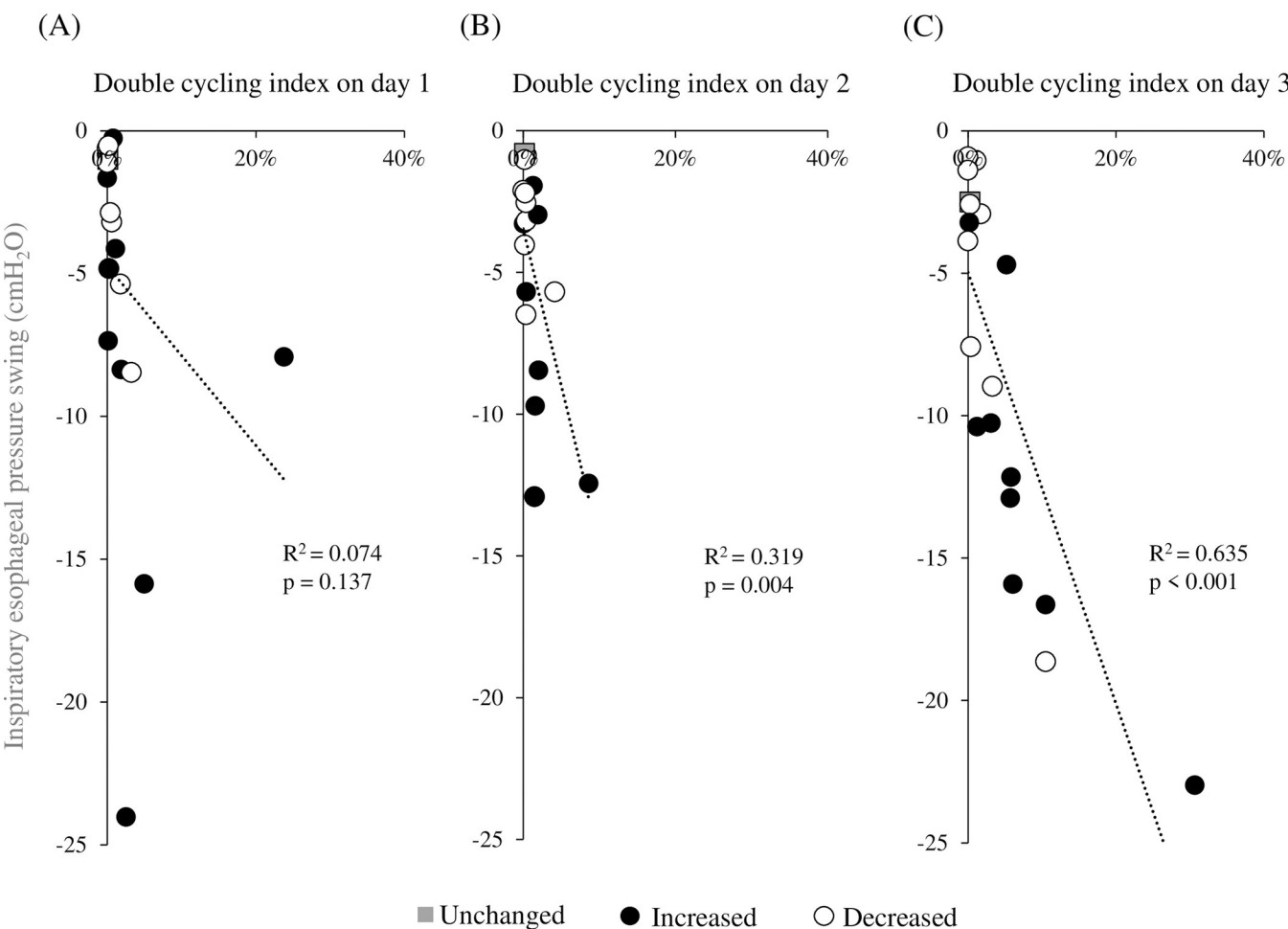

**Fig 4.** Correlation between the inspiratory esophageal pressure swing and the double cycling index on days 1 (A), 2 (B), and 3 (C).

mode of ventilation was used; thus, the impact of the type of mode was not precisely evaluated. In volume assist-control mode, tidal volume of double cycled breath is said to be higher than that of pressure assist-control mode, despite double cycling being more common in the latter [11]. Fourth, this study did not assess the double cycling caused by reverse triggering. Since, reverse triggering is commonly observed during the transition phase between deep sedation and the onset of patient triggering under assist-control ventilation [30], we eliminated the effect of disuse atrophy by accurately targeting patients with spontaneous breathing and assessed diaphragm thickness in a stable breathing cycle. Finally, to assess double cycling, 1-h offline breathing was applied in the breath evaluation. This period might be extremely short to obtain the representative values of each day. Thus, we conducted all measurements at same time every day to minimize variations caused by external conditions.

## Clinical implications

Strong breathing efforts cause P-SILI and concentric load-induced diaphragm injury in patients with severe ARDS [4, 20]. Thus, deep sedation or continuous NMBA infusion can be a reasonable option to prevent P-SILI [31]. However, there is a conflict between lung protection and diaphragm protection in terms of disuse atrophy of the diaphragm [21]. Thus, it is

important to identify patients who truly need to control their inspiratory effort from the perspective of both lung and diaphragm protection. $\Delta P_{es}$ is a major parameter of inspiratory effort, but its measurement requires the placement of a dedicated esophageal balloon catheter. Based on our findings, frequent double cycling can be a surrogate marker of spontaneous breathing that injures both the lungs and diaphragm.

## Conclusions

Double cycling on the third day of mechanical ventilation was associated with strong inspiratory efforts and, possibly, changes in diaphragm thickness. Hence, it can function as a surrogate indicator of diaphragm-injurious breathing pattern. Future studies, including double cycling caused by reverse triggering, should be performed to assesses the clinical effects of double cycling on diaphragm function.

## Supporting information

**S1 Checklist. STROBE statement—checklist of items that should be included in reports of observational studies.**
(PDF)

**S1 Data.**
(XLSX)

## Acknowledgments

The authors thank all staff of the ICU of the Tokushima University Hospital for their assistance with this study.

## Author Contributions

**Conceptualization:** Taiga Itagaki, Jun Oto.

**Data curation:** Taiga Itagaki, Yusuke Akimoto, Yuki Nakano, Yoshitoyo Ueno, Manabu Ishihara, Natsuki Tane, Yumiko Tsunano.

**Formal analysis:** Taiga Itagaki.

**Funding acquisition:** Taiga Itagaki.

**Investigation:** Taiga Itagaki, Jun Oto.

**Methodology:** Taiga Itagaki.

**Project administration:** Jun Oto.

**Supervision:** Jun Oto.

**Validation:** Yusuke Akimoto, Yuki Nakano, Yoshitoyo Ueno, Jun Oto.

**Writing – original draft:** Taiga Itagaki.

**Writing – review & editing:** Taiga Itagaki, Yusuke Akimoto, Yuki Nakano, Yoshitoyo Ueno, Manabu Ishihara, Natsuki Tane, Yumiko Tsunano, Jun Oto.

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
