## [Decision Letter · Decision Letter 0]

10 Jun 2022

PONE-D-21-30599Effects of double cycling on diaphragm thickness during the early phase of mechanical ventilation: A prospective observational studyPLOS ONE

Dear Dr. Itagaki,

Thank you for submitting your manuscript to PLOS ONE. After careful consideration, we feel that it has merit but does not fully meet PLOS ONE’s publication criteria as it currently stands. Therefore, we invite you to submit a revised version of the manuscript that addresses the points raised during the review process.

We look forward to receiving your revised manuscript.

Kind regards,

Steven Eric Wolf, MD

Academic Editor

PLOS ONE

Journal Requirements:

 “This work was partly supported by JSPS KAKENHI Grant Number JP21K16574 awarded to Dr. Taiga Itagaki. https://kaken.nii.ac.jp/grant/KAKENHI-PROJECT-21K16574/

Additional Editor Comments (if provided):

Editor - Thank you for submitting your paper to us for review. I sent it to eight distinguished referees for comment and decision of whom two agreed to review; you will see these below. They thought that the paper has merit, but each have raised some substantial issues to be addressed in a revision. Please carefully consider the comments below and reply directly to each in a cover letter with appropriate marked and linked changes to the manuscript. I look forward to seeing the revision, which I will send back to the same referees for further comment and decision. Please understand that this is not a guarantee of future publication, as the revised manuscript itself must stand on its own merit.

Reviewers' comments:

Reviewer's Responses to Questions

**Comments to the Author**

1. Is the manuscript technically sound, and do the data support the conclusions?

Reviewer #1: Yes

Reviewer #2: Yes

2. Has the statistical analysis been performed appropriately and rigorously? 

Reviewer #1: Yes

Reviewer #2: Yes

3. Have the authors made all data underlying the findings in their manuscript fully available?

Reviewer #1: Yes

Reviewer #2: No

4. Is the manuscript presented in an intelligible fashion and written in standard English?

Reviewer #1: Yes

Reviewer #2: Yes

5. Review Comments to the Author

Reviewer #1: I suggest to explore whether inspiratory effort and double cycling are independently associated with diaphragmatic thickness. Probably, a multivariable logistic regression may help to discern it.

I suggest to modify the title to avoid the suggestion of a direct effect of double cycling on diaphragm thickness, by something like “Relationships between high respiratory effort and double cycling with diaphragm thickness….”

Page 10. The issue of sedation needs more data demonstrating that sedation was reduced in “increased group”, whereas opioids were slightly higher. Patients with higher inspiratory effort commonly need more sedatives (except if excessive opioids induce higher respiratory effort in unconscious patients).

In the limitations section, I suggest to include the specific topic of your use of Pressure assist/control modes and whether the problem should be higher in case of using Volume assist/control modes.

Your “Clinical implications” section is not supported by your results. I suggest to focus it more on the interest of avoiding double cycling by ventilator settings adjustments

Reviewer #2: Itagaki and colleagues nicely attempt to establish a relation between an important asynchrony and diaphgram thickness. Although the text is clearly written and results presented coherently, I would suggest a few clarifications to improve the message of the manuscript.

1. Study rationale and hypothesis: although the background is stated it is not clear to the reader what is the author’s hypothesis being tested. What is the hypothesized role of double cycling on thickness? Please clarify the hypothesis you are testing, and especially how the timing is expected to influence (the different days).

2. Introduction: However, the effects of patient–ventilator asynchronies on VIDD have not been evaluated in clinically settings. So are you making a direct parallel between thickness and function?

3. Still on research question, this can be clarified: “Moreover, the role of respiratory

effort in the occurrence of double cycling during the early phase of mechanical

ventilation was evaluated”. What do you mean by role?

4. Methods: “On days 1, 2, 3, 5 and 7 after the initiation of mechanical ventilation, the

diaphragm thickness at peak inspiration and end-expiration (Tdiee) was examined via ultrasonography during spontaneous breathing”.

Please be more specific to allow repeatability of your measurements. What does during spontaneous breathing means, were the patients still assisted in A-PCV (not PSV correct?). I understand the echo measurements were detailed in a previous work, but important to detail how patients were breathing when you made the measurement.

5. P value for trend in Figure 2: can you specify how this was obtained. Simply comparing two timepoints or by analyzing the trend?

6. Please do not repeat data that was put in tables again in the text e.g for table 2.

7. Results for RASS score and sedation are presented which were not anticipated in the methods. Please adjust.

8. Analysis of the primary endpoint: you state that the primary endpoint of the study is the “correlation between the DC index and the maximum change in D. thickness on the first 3 days”. Please specify in the statistical analysis if you analyze days separately to avoid lumping repeated measures. In the statistical analysis paragraph is useful to have the analysis of the primary endpoint come first, not last.

9. Discussion: I find you should accompany better the reader to the relation between double cycling – effort and thickness, as now it’s pretty confusing.

E.g. you start by saying that “Considering that excessive inspiratory effort is a risk

factor of double cycling[12, 13], double cycling and diaphragm hypertrophy are both

associated with strong inspiratory efforts.”

This sentence is unclear, please rephrase. I would use the first paragraph to discuss the

primary outcome alone. What to grasp from the correlation between thickness and

double cycling? Then introduce the effort and how this interacts.

In addition the discussion is imbalanced, towards effort, while the primary aim is stated to be the study of double cycling. Is this a wanted feature?

6. PLOS authors have the option to publish the peer review history of their article (what does this mean?). If published, this will include your full peer review and any attached files.

Reviewer #1: **Yes: **Rafael Fernandez Fernandez

Reviewer #2: No

---

## [Author Response · Author response to Decision Letter 0]

8 Jul 2022

Journal Requirements:

[Response]

We ensured that our manuscript was meeting PLOS ONE’s style requirements, including those for file naming.

 “This work was partly supported by JSPS KAKENHI Grant Number JP21K16574 awarded to Dr. Taiga Itagaki. https://kaken.nii.ac.jp/grant/KAKENHI-PROJECT-21K16574/

[Response]

We amended our funding statement regarding to all support as you suggested. Also, we included the same statement within my cover letter.

[Response]

We uploaded our study’s minimal underlying data set as Supporting Information file.

 

Review Comments to the Author

Reviewer #1: 

1. I suggest to explore whether inspiratory effort and double cycling are independently associated with diaphragmatic thickness. Probably, a multivariable logistic regression may help to discern it.

[Response]

Thank you for your valuable comments. We understand that multivariable logistic regression is ideal for identifying an independent association between inspiratory effort/double cycling and diaphragm thickness. However, the sample size was too small for a multiple regression analysis. We have mentioned this point in the limitations section. 

2. I suggest to modify the title to avoid the suggestion of a direct effect of double cycling on diaphragm thickness, by something like “Relationships between high respiratory effort and double cycling with diaphragm thickness….”

[Response]

We agree with the idea to avoid asserting the direct effect of double cycling on diaphragm thickness. Accordingly, we have revised the title as “Relationships between double cycling and respiratory effort with diaphragm thickness during the early phase of mechanical ventilation: A prospective observational study.”

3. Page 10. The issue of sedation needs more data demonstrating that sedation was reduced in “increased group”, whereas opioids were slightly higher. Patients with higher inspiratory effort commonly need more sedatives (except if excessive opioids induce higher respiratory effort in unconscious patients).

[Response]

Because each patient received different sedatives, we do not have more data demonstrating that sedation was reduced among patients with increased diaphragm thickness. Nevertheless, sedation levels in the increased group had a significant tendency to be lighter over time, which may have led to increased inspiratory effort. Maintaining spontaneous breathing effort once the patient’s condition stabilizes may be the gold standard of mechanical ventilation. However, this strategy may have harmful effects on the diaphragm, such as load-induced diaphragm injury. Thus, we investigated the relationship between sedation levels and inspiratory effort on a daily basis. We have described this issue in the introduction and discussion. 

4. In the limitations section, I suggest to include the specific topic of your use of Pressure assist/control modes and whether the problem should be higher in case of using Volume-assist/control modes.

[Response]

Thank you for raising this very important issue. In the limitations section, we have added that we used only pressure-assist/control mode and that the results might be different if we used the volume-assist/control mode .

5. Your “Clinical implications” section is not supported by your results. I suggest to focus it more on the interest of avoiding double cycling by ventilator settings adjustments

[Response]

We reorganized the clinical implications section to be based on the results of this study. 

We understand that the management of double cycling by ventilator adjustments is important. However, we did not include this in the clinical implications because we believe that protecting the diaphragm from excessive inspiratory effort is of greater importance. 

 

Reviewer #2: 

Itagaki and colleagues nicely attempt to establish a relation between an important asynchrony and diaphragm thickness. Although the text is clearly written and results presented coherently, I would suggest a few clarifications to improve the message of the manuscript.

[Response]

Thank you for reviewing our paper. We have addressed all your concerns by providing a point-by-point response. We are certain that our manuscript has significantly improved thanks to your comments. We hope that you will now find the revised manuscript suitable for publication in PLOS ONE.

1. Study rationale and hypothesis: although the background is stated it is not clear to the reader what is the author’s hypothesis being tested. What is the hypothesized role of double cycling on thickness? Please clarify the hypothesis you are testing, and especially how the timing is expected to influence (the different days).

[Response]

We rephrased the introduction to strengthen the study rationale and clarified our hypothesis by stating that “we hypothesized that double cycling is associated with increased diaphragm thickness caused by strong inspiratory effort.” It was also our concern whether the increased inspiratory effort due to lightening sedation level to maintain spontaneous breathing once patient’s condition stabilizes (probably standard strategy) would have a negative effect on the diaphragm in some cases. Therefore, we also explained this research question in the introduction.

2. Introduction: However, the effects of patient–ventilator asynchronies on VIDD have not been evaluated in clinically settings. So are you making a direct parallel between thickness and function?

[Response]

Goligher et al. measured the thickening fraction of the diaphragm (TFdi) by ultrasonography and reported shorter ICU stay in patients with TFdi, equivalent to that of healthy subjects at rest (15%–30%) (ref. 7). Patients under mechanical ventilation can present with increased and decreased diaphragm thickness, which were both associated with prolonged ventilator dependence and a higher mortality rate. Thus, in our study, sudden changes in diaphragm thickness were considered as a form of dysfunction. 

3. Still on research question, this can be clarified: “Moreover, the role of respiratory effort in the occurrence of double cycling during the early phase of mechanical ventilation was evaluated”. What do you mean by role?

[Response]

We apologize for our poor phrasing. We explored the involvement of inspiratory effort in the occurrence of double cycling. We have reworded this accordingly as “involvement” instead of “role.”

4. Methods: “On days 1, 2, 3, 5 and 7 after the initiation of mechanical ventilation, the diaphragm thickness at peak inspiration and end-expiration (Tdiee) was examined via ultrasonography during spontaneous breathing”. Please be more specific to allow repeatability of your measurements. What does during spontaneous breathing means, were the patients still assisted in A-PCV (not PSV correct?). I understand the echo measurements were detailed in a previous work, but important to detail how patients were breathing when you made the measurement.

[Response]

We clarified the condition which we assumed as spontaneous breathing as follows: “On days 1, 2, and 3, esophageal pressure (Pes) was examined; the patient was assumed to be spontaneously breathing when the deflection of Pes was initiated before the rise of airway pressure (Paw). On days 3 and 7, the patient was assumed to be spontaneously breathing when actual respiratory frequency was greater than the set value for respiratory rate.”

5. P value for trend in Figure 2: can you specify how this was obtained. Simply comparing two timepoints or by analyzing the trend?

[Response]

There was a mistake in description of ANOVA used. We performed one-way ANOVA with multiple comparisons to compare the effect of time (days) on the mean change in end-expiratory diaphragm thickness, double cycling index, ΔPes, thickening fraction, RASS, and fentanyl use. Thus, the p-value in Figure 2 represents statistically significant differences between at least two time points. We collected the type of ANOVA used in statistical analysis section and clarified this in figure legend of Figure 2.

6. Please do not repeat data that was put in tables again in the text e.g for table 2.

[Response]

We removed data repeatedly written in the text that was already put in table 2.

7. Results for RASS score and sedation are presented which were not anticipated in the methods. Please adjust.

[Response]

In addition to the description of the assessment of the time-weighted Richmond Agitation–Sedation Scale (RASS) score and fentanyl use on days 1, 2, and 3 in the methods, we included the effect of sedation levels on changes in inspiratory effort as secondary endpoints.

8. Analysis of the primary endpoint: you state that the primary endpoint of the study is the “correlation between the DC index and the maximum change in D. thickness on the first 3 days”. Please specify in the statistical analysis if you analyze days separately to avoid lumping repeated measures. In the statistical analysis paragraph is useful to have the analysis of the primary endpoint come first, not last.

[Response]

We performed one-way ANOVA with multiple comparisons, and we have corrected this. We reconstructed the statistical analysis section so that the primary endpoint is mentioned first. 

9. Discussion: I find you should accompany better the reader to the relation between double cycling – effort and thickness, as now it’s pretty confusing. E.g. you start by saying that “Considering that excessive inspiratory effort is a risk factor of double cycling [12, 13], double cycling and diaphragm hypertrophy are both associated with strong inspiratory efforts.” This sentence is unclear, please rephrase. I would use the first paragraph to discuss the primary outcome alone. What to grasp from the correlation between thickness and double cycling? Then introduce the effort and how this interacts.

[Response]

Thank you for the suggestion. Indeed, we tried to discuss the primary outcome alone. However, since our hypothesis was double cycling is associated with increased diaphragm thickness caused by strong inspiratory effort, it was difficult to discuss the relationship between double cycling and the diaphragm aside from inspiratory effort. Therefore, we referred to factors other than inspiratory effort as follows: “However, eccentric contraction of the diaphragm, a possible mechanism by which a certain type of asynchronies affects the diaphragm[8], and the characteristics of double cycling itself, such as increased tidal volume[11], may have still influenced diaphragm injury. Further studies are needed to clarify this issue.” Afterwards, we discussed the effort related to both double cycling and diaphragm injury, rephrasing the sentence you pointed out.

10. In addition, the discussion is imbalanced, towards effort, while the primary aim is stated to be the study of double cycling. Is this a wanted feature?

[Response]

In this revised manuscript, we clearly stated our hypothesis that double cycling is associated with increased diaphragm thickness due to strong inspiratory effort. The main message of this study is the frequent double cycling may be a surrogate marker of spontaneous breathing, which damages both the diaphragm and the lungs. We have discussed this in the clinical implications section.

---

## [Decision Letter · Decision Letter 1]

4 Aug 2022

Relationships between double cycling and respiratory effort with diaphragm thickness during the early phase of mechanical ventilation: A prospective observational study

PONE-D-21-30599R1

Dear Dr. Itagaki,

We’re pleased to inform you that your manuscript has been judged scientifically suitable for publication and will be formally accepted for publication once it meets all outstanding technical requirements.

Kind regards,

Steven Eric Wolf, MD

Academic Editor

PLOS ONE

Additional Editor Comments (optional):

Reviewers' comments:

Reviewer's Responses to Questions

**Comments to the Author**

1. If the authors have adequately addressed your comments raised in a previous round of review and you feel that this manuscript is now acceptable for publication, you may indicate that here to bypass the “Comments to the Author” section, enter your conflict of interest statement in the “Confidential to Editor” section, and submit your "Accept" recommendation.

Reviewer #1: All comments have been addressed

2. Is the manuscript technically sound, and do the data support the conclusions?

Reviewer #1: (No Response)

3. Has the statistical analysis been performed appropriately and rigorously? 

Reviewer #1: (No Response)

4. Have the authors made all data underlying the findings in their manuscript fully available?

Reviewer #1: (No Response)

5. Is the manuscript presented in an intelligible fashion and written in standard English?

Reviewer #1: (No Response)

6. Review Comments to the Author

Reviewer #1: (No Response)

7. PLOS authors have the option to publish the peer review history of their article (what does this mean?). If published, this will include your full peer review and any attached files.

Reviewer #1: **Yes: **RAFAEL FERNANDEZ

---

## [Editor Report · Acceptance letter]

8 Aug 2022

PONE-D-21-30599R1 

Relationships between double cycling and inspiratory effort with diaphragm thickness during the early phase of mechanical ventilation: A prospective observational study 

Dear Dr. Itagaki:

I'm pleased to inform you that your manuscript has been deemed suitable for publication in PLOS ONE. Congratulations! Your manuscript is now with our production department. 

Kind regards, 

on behalf of

Dr. Steven Eric Wolf 

Academic Editor

PLOS ONE